# Children's Rights by Design and Internet Governance: Revisiting General Comment No. 25 (2021) on Children's Rights in Relation to the Digital Environment

## Christian Djeffal

TUM School of Social Sciences and Technology, Technical University of Munich, 80333 Munich, Germany; christian.djeffal@tum.de

**Abstract:** This article reviews how children's rights can be considered a driver of internet governance based on general comment No. 25 (2021) on children's rights in relation to the digital environment. This instrument translates the rights derived from the UN Convention on the Rights of the Child from the pre-digital era to the application of children's rights to current issues of digitization. In the introduction, I explain how this general comment was drawn up and what its legal significance is. This article briefly summarizes the content of the general comment and then goes on to discuss the main impacts of this instrument on internet governance, namely, substantial shifts, children's rights by design, the law's binding nature, and participation.

**Keywords:** children's rights; internet governance; human rights; law by design; digitization; content moderation

## 1. Introduction

On 2 March 2021, the Committee on the Rights of the Child released General Comment No. 25 (2021) on children's rights in relation to the digital environment (General Comment 25), to inform states on how to uphold the human rights of children in a digital environment. These human rights are mainly enshrined in the UN Convention on the Rights of the Child (UNCRC) (United Nations General Assembly 1989, p. 3), which has 196 member states.

This convention, which was adopted on 20 November 1989, stems from the pre-internet era. The question of how to protect children's rights online has been the subject of discussions for a long time (Wilson and McAloney 2010; Hasse et al. 2019; Lievens 2010; Nyamutata 2019; Sonia et al. 2016). The Committee on the Rights of the Child addressed the internet on several occasions, such as addressing that children might be exposed to offensive material in early childhood (Committee on the Rights of the Child 2005, para. 35), the access to appropriate information for disabled children (Committee on the Rights of the Child 2006, para. 37), cyberbullying (Committee on the Rights of the Child 2011, para. 21), sexual abuse via information and communication technologies (Committee on the Rights of the Child 2011, para. 30), and internet addiction (Committee on the Rights of the Child 2013, para. 38); however, it has never given focused attention to digital technologies in a cross-cutting manner to update the UNCRC. The question regarding how children's rights can be updated feeds into the larger question of how international law changes over time (Djeffal 2016, p. 28). This question is constantly on the table since international treaties remain unchanged for long periods, as changes often require unanimity and involve many parties.

In the case of human rights treaties, one effective method is to use general comments provided by human rights committees. In such comments, the competent committees explain how human rights law is to be applied in a specific context or to a specific question (Ando 2012). The Committee on the Rights of the Child drew "on the Committee's experience in reviewing States parties' reports, its day of general discussion on digital media and children's rights, the jurisprudence of the human rights treaty bodies, the recommendations

of the Human Rights Council and the special procedures of the Council, two rounds of consultations with States, experts and other stakeholders on the concept note and advanced draft and an international consultation with 709 children living in a wide variety of circumstances in 28 countries in several regions" (Committee on the Rights of the Child 2021, para. 5). The Committee released an invitation to all interested stakeholders to present their inputs in March 2019 and August 2020 and received numerous answers from states, regional organizations and United Nations Agencies, national human rights agencies and children's commissioners, children's and adolescent's groups, and other stakeholders.

## 2. What the General Comment Says

General Comment 25 is structured in 14 sections with a specific methodology.[1] After an introduction, the general purpose of the document is stated as follows:

> In the present general comment, the Committee explains how States parties should implement the Convention in relation to the digital environment and provides guidance on relevant legislative, policy and other measures to ensure full compliance with their obligations under the Convention and the Optional Protocols thereto in the light of the opportunities, risks and challenges in promoting, respecting, protecting and fulfilling all children's rights in the digital environment. (Committee on the Rights of the Child 2021, para. 7)

To achieve this goal, the general comment first restates the essential general tenets of children's rights, which are the general principles of children's rights (III) and their evolving capacities regarding autonomy (IV). The principles are basically a re-iteration of the important standards of the rights of the child, including non-discrimination (A); the best interests of the child (B); the right to life, survival, and development (C); and respect for the views of the child (D), while the concept of evolving capacities entails the vulnerability of children and the fact that their autonomy gradually evolves.

The general comment then explains the general measures of implementation (V) and later comments on the related governance issues of special protection measures (XII), international and regional cooperation (XIII), and dissemination (XIV). All of these parts show that this document has a wide scope that goes beyond legal measures. The general comment also states that "[t]he cross-border and transnational nature of the digital environment necessitates strong international and regional cooperation, to ensure that all stakeholders, including States, businesses and other actors, effectively respect, protect and fulfil children's rights in relation to the digital environment." This shows that the instrument also takes into account a wide array of stakeholders including civil society (V. H.) and the business sector (V. I.).

Regarding the material questions of children's rights online, the general comment outlines six areas of children's rights, which are investigated more substantively:

- Civil rights and freedoms (VI);
- Violence against children (VII);
- Family environment and alternative care (VIII);
- Children with disabilities (IX);
- Health and welfare (X);
- Education, leisure, and cultural activities (XI).

This section comes closest to an actual commentary of different articles in the UNCRC. It highlights several problems and interferences with children's rights and points to possible routes for the mitigation of the respective risks. To give just one example, the section on the freedom of expression (IV. B.) highlights issues concerning the freedom of expression online. It starts from the observation that children find opportunities to express themselves online (Committee on the Rights of the Child 2021, para. 58) and then goes on to state that measures, such as filters, must be lawful, necessary, and proportionate and that children

---

[1] The following references in brackets refer to the numbering according to the structure of General Comment 25.

should receive information and training to exercise their rights (The National 2019). It also stresses the active obligations of states to protect children from third-party interference where they make use of their freedom of expression, and mentions "cyberaggression, and threats, censorship, data breaches and digital surveillance" (Committee on the Rights of the Child 2021, para. 60). It also touches upon the new affordances of artificial intelligence and calls upon states to "ensure that uses of automated processes of information filtering, profiling, marketing and decision-making do not supplant, manipulate or interfere with children's ability to form and express their opinions in the digital environment" (ibid., para. 61). This shows how General Comment 25 touches upon different questions of the online environment and looks at the negative and positive functions of human rights.

### 3. Impulses of International Law for Internet Governance

While international law and internet governance are sometimes treated as mutually exclusive means by which to steer the internet, I argue that they have a large overlap. International law deals with internet governance issues in numerous regards. Consequently, international law contains an important set of instruments for internet governance. The fact that the concept of governance has broadened the view on instruments other than the law does not mean that law does not form part of internet governance. As Rolf Weber showed, there is a significant discussion regarding new forms of regulation to tackle major issues of internet governance (Weber 2021, p. 13). This and other instances exemplify how internet governance has sparked new thinking in international law. However, this Special Issue shifted the focus regarding the use of international law toward internet governance. This article argues that children's rights clearly bring to light certain aspects of the potential of international law to affect internet governance.

### 3.1. Substantial Shifts

My first claim is that the regulatory technique used by the UNCRC and General Comment 25 highlights the emerging nature of digital technologies and the fact that they circle back to the offline world. This underscores a general conundrum in that internet governance has to deal with several substantial sociotechnical shifts (Blumenthal and Clark 2001, p. 70). One significant challenge of internet governance is to accommodate several emerging technologies that affect the internet but also have impacts when they are not used in a networked setting. Take, for example, artificial intelligence (AI). AI systems are used for content moderation online and targeted advertising, but have many applications beyond this (Djeffal 2020a, pp. 280–82). In important fora for internet governance, such as the IGF, there are many sessions on other emerging technologies like AI, distributed ledger technologies, and, recently, brain–computer interfacing and quantum computing. How can one accommodate all of these technologies from the perspective of internet governance? While in the last twenty years, many commentators on internet governance have struggled substantively to understand this as its own "field" or discourse (DeNardis 2013; Mueller and Badiei 2020), the time might be ripe to think about how to effectively link internet governance with other "fields" or discourses.

One way to do so is to shift the center of attention from specific technologies to material questions that transgress the limits of certain technologies. Human rights, and, more specifically, children's rights, are such focal points that help to keep track of different technologies. Thus, perhaps one way to affect internet governance from the perspective of international law would be to single out important focal points and to determine the links to other issue areas from this perspective. General Comment 25 shows how it is possible to uphold something that could or should be considered a core value of the internet in different digital settings that do not necessarily link to children's rights. This could have several advantages for internet governance. First, internet governance and its rich discourse could be translated to other settings and gain more relevance in other fields and discourses. Public international law and, specifically, human rights law would be a tool for the translation of the findings of internet governance. Second, this would also allow

the field of internet governance to import information and findings from other discourses to be up to date, as it cannot be expected that internet governance fora will lead on every single topic. Third, from a bird's-eye perspective, several discourses can be coupled and linked through specific issues, which might enable them to be synchronized but could also create ruptures.

Another context in which international law might become more important is a spillover situation into offline settings. While the internet has traditionally affected a very limited set of considerations, an analysis from a human rights perspective shows how multifaceted the effects of the internet are today. Sections VI–XI of the General Comment provide evidence for the many connections to civil, political, social, and cultural rights. This is an expression of the general-purpose nature of the internet, but it is also a constant challenge for internet governance to track its uses. One specific example of these increasing effects of the internet is the repercussions on the "offline world", which can be gleaned from General Comment 25 in several instances. The section on violence (VII) maps the different ways in which the digital environment also leads to children becoming victims of violence in the real world and mentions examples such as sexual offenders using social media (Committee on the Rights of the Child 2021, para. 81) or the availability of information and communication that sparks children to inflict harm to themselves (ibid., para. 80). The strategic use of social media has led to numerous discussions, as such offenders can get to know children and gain their trust through the affordances of content moderation algorithms by using mobile devices near schools or other places in which they expect children to be (Committee on the Rights of the Child 2021, para. 88). The trust that child offenders gain online enables them to violate and exploit children in the real world. In tragic cases, the sharing of pictures and videos of these crimes perpetuates the violations again online. A human-rights-focused analysis can capture several switches between online and offline activities. The Molly Russel case is a tragic example of how online behavior links to self-harm offline. Molly Russel took her life days before her fourteenth birthday (The National 2019). It was revealed that she was bombarded by suicidal content online. This led to a policy discussion in the United Kingdom that finally resulted in the proposal for an online safety bill (Department for Digital Culture Media and Sport and Nadine Dorries 2022). Both of these cases have been discussed in connection to children's rights, which shows that human rights and international law can both serve well to track the impacts, irrespective of whether they occur online or offline and irrespective of the intersections.

### 3.2. Children's Rights by Design

My second claim is that General Comment 25 can be read as a collection of children's-rights-by-design norms, which reinforces governance through law-by-design. This could inform internet governance regarding how to think about impacts on technology design, especially in the context of innovations, but to do so from a normative grounding that focuses on vulnerability and human rights rather than on technologies. Thinking about more legal design norms apart from privacy, IT-security, and general references to human rights could be an important instrument for internet governance in the future. This requires one to reflect on what law-by-design norms actually are. Such by-design thinking has traditionally been applied in the area of privacy and data protection (Cavoukian 2009; Schartum 2016; Domingo-Ferrer et al. 2014; Resolution on Privacy by Design 2010), as well as IT security (Bygrave 2022), but is now considered to have spread throughout the legal system (Djeffal 2020b, p. 857). This has gained traction in internet governance through several human rights discourses in the context of the governance of internet protocols, especially through the IEFT or in the context of the Domain Name System (DNS) (Mueller and Badiei 2019, p. 61). Even if they are linked to legal frameworks, such as human rights, these efforts are regularly structured around certain technical or sociotechnical topics, such as internet protocols or the DNS. Part of the potential of legal governance mirrored in General Comment 25 is to complement this view with a clear normative focus that works across technologies and their sociotechnical settings, which focuses more on a set of norms as opposed to a technical

setting. This introduces a new perspective that can inform different topics of technology governance. Law-by-design norms profit from the law's binding nature and combine it with normative claims that are to be translated into technology. The law's potential for enforcement gives strength to general notions calling for the early inclusion of societal values into technology development processes. It thereby reinforces general concepts such as responsible research and innovation (Stilgoe and Guston 2017). Legal governance also brings specific socio-technical aspects of the design to the forefront. This means that it is often not only the technology itself that is designed but also its surroundings, including organizational and procedural aspects. While this is included in the very notion of governance, the legal determination of these issues makes them visible and debatable and is, therefore, a precursor of democratic governance.

General Comment 25 entails several design elements and thereby extends the design focus to children's rights in general. It prescribes in several instances the necessity and the pathways by which to achieve children's rights in the sociotechnical design of technology. While privacy by design is a consolidated category, General Comment 25 shows that design obligations have proliferated substantially in international law.

The very core of the problem is related to the fact that "[t]he digital environment was not originally designed for children", which means that one of the responsibilities of states is to further redesign the digital environment today and ensure a child-appropriate design going forward (Committee on the Rights of the Child 2021, para. 12). General Comment 25 mentions rather traditional design goals, such as data protection and privacy by design and cybersecurity and safety by design. However, other sections stress design goals that are less prominent in other contexts. One interesting approach in this regard is to situate design for children in a more general context of universal design that also includes other groups outside of the norm of the standard user.[2] These design goals can play a role in the different uses of technologies, including content recommendations in the context of social media (Committee on the Rights of the Child 2021, para. 37), and in particular, technologies to profile children (ibid., para. 62), games, and digital play (ibid., para. 108). An important aspect is also to design explanations in an age-appropriate manner (ibid., para. 39).

Impacts on design processes form an important part of the measures that states should take. Comprehensive policies also include, apart from regulation, "industry codes and design standards" (Committee on the Rights of the Child 2021, para. 24). This means that it is necessary to extend the governance to the business sector, where many, if not most, of the relevant systems are designed. General Comment 25 specifically mentions several measures that impact the design process, e.g., to "undertake child rights due diligence, in particular to carry out child rights impact assessments and disclose them to the public, with special consideration given to the differentiated and, at times, severe impacts of the digital environment on children" (ibid, para. 38). This section shows not only that it is important to understand the general impacts but also to understand that the effects can be differentiated and that they should be discussed publicly. Furthermore, "States parties should require all businesses that affect children's rights in relation to the digital environment to implement regulatory frameworks, industry codes and terms of service that adhere to the highest standards of ethics, privacy and safety in relation to the design, engineering, development, operation, distribution and marketing of their products and services" (ibid, para. 39). Such obligations link regulation to technology design and help to operationalize children's rights on the ground. The drafters of the regulatory framework are aware that this will mean prioritizing children's rights over the commercial interests of the respective companies and deliberately mentioning design obligations in this regard.

One aspect that also relates to measures is furthering innovation to achieve design goals that are established in the context of privacy-enhancing technologies (PETs) (Hes and Borking 2000). An example of an approach that explicitly focuses on public inter-

---

2    Para. 110 reads: "By introducing or using data protection, privacy-by-design and safety-by design approaches and other regulatory measures, States parties should ensure that businesses do not target children using those or other techniques designed to prioritize commercial interests over those of the child."

est technology development through innovations can be found in the context of children, which requires states to "promote technological innovations that meet the requirements of children with different types of disabilities and ensure that digital products and services are designed for universal accessibility so that they can be used by all children without exception and without the need for adaptation" (Committee on the Rights of the Child 2021, para. 39). This obligation can be linked to the UN Convention on the Rights of Persons with Disabilities, which contains a technology innovation clause in Art. 4 g (Broderick 2018). Another example is the encouragement of "innovation in digital play and related activities that support children's autonomy, personal development and enjoyment" (Committee on the Rights of the Child 2021, para. 108).

### 3.3. Process: Ways to Achieve Binding Results

General Comment 25 can also inform internet governance on how to formalize norms and informal behavior. In analyses of internet governance, the general distinction between formal international law and informal norms is still the outset of many reflections (Eichensehr 2014, p. 346; Lehto 2021). This is certainly a correct way of conceptualizing the situation. However, Comment 25 sheds light on how norms and behavior can be formalized in settings in which this is considered necessary. In multi-stakeholder settings, there it has been discussed at great length how to turn results into actual policies. As a particular example, the Internet Governance Forum has been struggling for years to turn agreements into actionable outcomes. When the protection of vulnerable groups, such as children, is at stake, it is often not satisfactory to wait for the behavior of the relevant stakeholders to improve to an acceptable level. While the law is certainly not the only means of internet governance, one of the law's advantages is the fact that it aspires to be binding and includes several routes to adjudicate problems and enforce the respective results. One important specific aspect of international law is that law creation is not only based on treaty-making at large conferences; there are more subtle means to turn the behavior of several actors into law.

Their legal significance depends, on the one hand, on the degree to which the respective committee is empowered to render authoritative interpretations of the treaty. On the other hand, given that many committees reviewed the state reports, the normative relevance can also stem from subsequent practice, as foreseen in Art. 31 (3) (b) of the Vienna Convention on the Law of Treaties (1969) and discussed by Georg Nolte (2013). The Committee on the Rights of the Child has no express authority to issue general comments; however, the Committee has included this in Rule 73 of its Provisional Rules of Procedure (Committee on the Rights of the Child 2003, p. 21). The competence to do so is based on implied powers regarding the task of recognizing state reports according to Art. 44 Sec. 1 CRC to make suggestions and recommendations pursuant to Arts. 44 and 45. There is also the competence of communicating this to the state parties according to Art. 45 (d) to "make suggestions and general recommendations based on information received pursuant to Arts. 44 and 45 of the present Convention. Such suggestions and general recommendations shall be transmitted to any State Party concerned and reported to the General Assembly, together with comments, if any, from States Parties." The wording of this norm suggests that the general comments are prima facie non-binding suggestions, and recommendations lack the element of a binding and cogent utterance (Evans 2020, p. 536). However, the normative force can also be a result of the process of states accepting such recommendations via their comments and through acknowledgment in the United Nations General Assembly. Therefore, general comments are an important part of communication that can lead to binding results that are reinforced by human rights law.

Currently, the contents of General Comment 25 are considered by the Third Committee of the UN General Assembly.[3] This process could result in a confirmation of General

---

[3]   See, for example, A/C.3/76/L.25/Rev.1 Seventy-sixth session Third Committee Agenda item 70: (a) Promotion and protection of the rights of children, 11 November 2021.

Comment 25 through the subsequent practice of the states according to Art. 31 Sec. 3 Subsec. (b) VCLT. Even in cases in which the interpretation of the Committee on the Rights of the Child would have to be considered an evolutive and dynamic interpretation, this could be justified by the practice of the states. Interpretations by the committee might also be based on the object and purpose as provided for in Art. 31 Sec. 1 VCLT. Both techniques of treaty interpretation can also be combined. These mechanisms show how international law builds bridges to binding and enforceable results while constantly considering stakeholder views. This participatory approach is even more important when it comes to the inclusion of children as key stakeholders.

*3.4. Participation: Inclusion of Stakeholders of Varying Capacities*

Multistakeholderism, which has been one of the central tenets of internet governance, relies on the participation of the affected groups (Hofmann 2016). However, multistakeholderism was criticized for turning a blind eye toward vulnerable groups (Sonia et al. 2016, pp. 12–14). Children's rights not only include obligations regarding the participation of children, it has been argued that participation is a central function of children's rights that complements the functions of respecting and protecting children's rights, which have been termed provision, protection, and participation (Heimer et al. 2018), respectively. This is a good example of a stakeholder group that cannot easily make its voice heard but needs special attention and particular offers for participation (Malcolm 2015).

At least three main dimensions can be identified when it comes to the participation of children and their digital environment: first, participatory processes that directly relate to internet governance; second, access to internet-related activities; and third, access to real world activities that is mitigated through the internet. This broad understanding of participation also includes access to important aspects of societal life, which, in many instances, is a prerequisite for engagement. This is the case when children are directly included in governance issues. As previously mentioned, General Comment 25 relied on consultations that were carried out with 709 children. The General Comment stresses that "[c]hildren with disabilities should be involved in the design and delivery of policies, products and services that affect the realization of their rights in the digital environment" (Committee on the Rights of the Child 2021, para. 91). In many instances, it is emphasized that children ought to have access to the internet to participate in society. Therefore, children should have the "right to participate in organizations that operate partially or exclusively in the digital environment" (ibid., para. 65). It is also specifically mentioned that "[p]articipation in cultural life online contributes to creativity, identity, social cohesiveness and cultural diversity", and therefore, "States parties should ensure that children have the opportunity to use their free time to experiment with information and communications technologies, express themselves and participate in cultural life online" (ibid., para. 106). General Comment 25 also stresses that online activity can mediate participation in offline activities. Areas that are particularly stressed include political self-determination, participation in political processes (Committee on the Rights of the Child 2021, para. 16), and access to health services (ibid., para. 93) and education (ibid., para. 99). Again, this demonstrates that the online environment enables participation in both online and offline life.

**4. Conclusions**

As the concept of internet governance has matured, it is important to revisit the relationship between internet governance and international law with an integrative mindset that is less focused on conceptual differences and more on the potential for synergies. The development of children's rights shows that international legal instruments can be important measures in the quest to govern the internet. This contribution has focused on four main areas in which the evolution of children's rights reinforced internet governance. The first concerns the substantial shifts caused by the nature of emerging technologies and the ever-increasing number of technologies, which has led to calls for a constant reassessment of the impacts of a networked society. One example is the increasing merger of

online and offline realities. Second, the law has furthered by-design obligations that are relevant across technologies and shifted the focus to specific sets of norms, such as children's rights. Third, international law has developed mechanisms to turn social practices into binding legal norms. This can become an important tool that is used to support the central claims of internet governance. Fourth, children's rights are a good example of different forms of participation in political processes concerning internet governance, as well as participation in online activities or participation through activities online. Here, international law reinforces the goals of internet governance by allowing the steering of the digital environment toward a sustainable path for today's and tomorrow's generations.

**Funding:** This research received no external funding.

**Acknowledgments:** I acknowledge technical and research support by Daan Herpers and Salma Atallah.

**Conflicts of Interest:** The author declares no conflict of interest.

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
