# Peer review of "Children’s Rights by Design and Internet Governance: Revisiting General Comment No. 25 (2021) on Children’s Rights in Relation to the Digital Environment"

_laws, 2021_

Round 1

Reviewer 1 Report

The author analyzes in the article current topic, related to child rights and internet governance. His statements are clear and I have no suggestions for author. 

Author Response

Thank you very much for the positive feedback. You have exactly grasped my idea!

Reviewer 2 Report

This piece has a good conceptual premise - it asks whether rights based approaches such as that demonstrated in General Comment No.25 (2021) offers a better way of regulating the digital environment than design of technologies approaches. The author suggests that the Comment might be a good template for a regulatory approach that based on a set of normative values. 

However, the piece is not very evaluative of either the General Comment or the state of the art in relation to prevailing ideas in internet governance. It describes the general comment and then gives a very patchy picture of the key debates on this issue in the realm of internet governance. There is no really evaluation of the general comment and the author does not articulate what it adds to the general debate beyond the idea that a values approach might be better. 

The structure of the piece really needs work to bring out these ideas. I suggest that the author outlines the current debate in internet governance outlining the arguments for a digital setting approach and a normative value approach, then outline what the General Comment adds to this this debate and then offer some kind of evaluation of the General Comment. Will it work? Are there any obvious weaknesses in it from internet governance point of view? The article should conclude by clearly articulating the significance of the General Comment from an internet governance perspective. 

Author Response

Thank you very much for the helpful comments and suggestions. As you can see from the version in track changes, I have re-drafted the whole article in order to improve it in accordance with your suggestions. I particularly took the comment of the reviewer to heart to make ideas clearer and to show exactly where an analysis of the General Comment No. 25 might include impulses for internet governance. I particularly value the interesting idea developed by the reviewer to change perspective and research question of the article. I know that the road I have taken is less common as international law is mostly analysed from the perspective of its insufficiencies, which have sparked major innovations in internet governance. However, I respectfully submit that after the field of internet governance has matured, it is promising to reconsider it from the perspective of international law. This also ties in well with the general theme of the special issue. I agree that there are limits to my inquiry, especially I cannot evaluate and assess GC 25 in detail. Yet, I aimed to oblige with the formal requirements of MDPI and I am convinced that this perspective can still complement the discussions. Therefore, I would be very grateful if the reviewer would acknowledge the many efforts I have made to improve the text according to his or her wishes.

Reviewer 3 Report

The subject of the paper is undoubtedly interesting. The perspective identified by the author - the relationship between the international protection of the child rights and internet governance - certainly deserves attention.

That said, it must be underlined that in the paper and, particularly, in the third paragraph, the explanation of the way in which the General Comment n. 25 should influence the development of internet governance does not appear sufficiently clear. The frequent use of expression like “law by design” does not help in the understanding of why the General Comment n. 25 should be a novelty in the development of internet governance, neither with respect to other instruments of international law that already deal with the topic of internet governance (thinks, others, think to data protection, cybersecurity, commerce), nor with respect to other instruments of protection of the rights of child (both, at international and regional level) which also contemplate the internet dimension. Moreover, the author does not take in consideration previous General Comments published by the CRC that already deal with the topic of the internet.

Al being considered, in order to be published, the paper should be reviewed clarifying the aspects above mentioned. Finally, I would suggest an in-depth analysis of the international law studies on internet regulation since the bibliography does not appear sufficiently complete.

Author Response

Thank you very much for the general interest in this article, the helpful comments and suggestions. As you can see from the version in track changes, I have re-drafted the whole article in order to improve it in accordance with the suggestions. I particularly took the comment of the reviewer to heart to make ideas clearer and to show exactly where an analysis of the General Comment No. 25 might include impulses for internet governance. Summaries are now contain in the beginning of each section of the text. I have attempted to clarify the idea of law by design and how it scales by being included on a broad front such as in GC 25. I strengthened the references to other areas of law by design, particularly to data protection and IT-security and included further references. I included also more studies on internet regulation as well as previous general comments on the internet. 

Round 2

Reviewer 2 Report

I think this has greatly improved. It is much more analytical rather than merely descriptive.